# The Concept of Moving School and Its Practical Implementation in Bavarian Higher Secondary Schools

**DOI:** 10.3390/children10081395

**Published:** 2023-08-16

**Authors:** Paul Englert, Christian Andrä, Yolanda Demetriou

**Affiliations:** 1Department of Sport and Health Sciences, Technical University of Munich, 80333 Munich, Germany; 2Department of Movement and Sport Pedagogy, University of Applied Sciences for Sport and Management, 14471 Potsdam, Germany; andrae@fhsmp.de; 3Department of Sport Science, University of Innsbruck, 6020 Innsbruck, Austria; yolanda.demetriou@uibk.ac.at

**Keywords:** Moving School, physical activity, curriculum analysis, school programmes, school homepages, Bavarian higher secondary schools

## Abstract

The roles of physical activity and the reduction in sedentary activities in the healthy physical, psychosocial and mental development of children and adolescents are undisputed. This is where the concept of Moving School comes in, which has been expanded to a holistic approach that takes into account all areas of the school, including lessons, after-school care and breaks, and aims to provide students with a sufficient level of physical activity. There are no current studies that show to what extent this concept has arrived in the theoretical specifications for higher secondary schools and to what extent it is implemented in practice. In this study, by conducting a narrative review, we determine the core elements of the concept of Moving School. Furthermore, we analyse the extent to which these elements of Moving School are taken into account in the school curriculum, in published school programmes and on school homepages. In this study, we revealed that the concept of Moving School has hardly been implemented in practice in Bavarian higher secondary schools as mostly only single elements of it are referred to in the school curriculum, programmes and school homepages. It can therefore be assumed that the concept has not yet been able to achieve its intended effect, namely, to fill the daily lives of schoolchildren with movement, play and sport.

## 1. Introduction

At higher secondary schools in Bavaria, all-day programmes have been further expanded in recent years. As a result, an increasing number of children and adolescents spend more time in schools [1]. At the same time, Bavarian higher secondary schools are making the transition from eight-year to nine-year schools. For this purpose, a new curriculum, the LehrplanPLUS, was developed and successively introduced in the 2017/2018 school year. The redesign of higher secondary schools with an additional grade level and the renewal of the curriculum can offer great opportunities to put previous structures to the test and renew them. At the same time, however, there is a growing responsibility for schools, where students spend much of their daytime, to provide them with a comprehensive educational experience that includes movement-related activities.

The roles of physical activity and the reduction in sedentary activities in the healthy physical, psychosocial and mental development of children and adolescents are undisputed [2,3]. The WHO recommends a minimum amount of moderate to vigorous physical activity of 60 min per day on average during a week for three-year-old to seventeen-year-old children and adolescents [4]. It is assumed that, with an increase in physical activity, the health benefits also increase until a certain degree [2,5,6]. In addition to the positive effects of exercise, the adverse effects of sedentary behaviour are becoming increasingly apparent. Since it has been proven that physical activity can compensate for the negative effects of sitting too long [7], schools should work all the harder to increase periods of physical activity and reduce sitting time.

The results of recent studies show that this has not yet been achieved. The second wave of the KiGGS study showed that, in Germany, only 22.4% of girls and 29.4% of boys in the 3–17 age group are physically active for at least 60 min per day, thus meeting the WHO recommendations. In addition, the prevalence of the recommended level of physical activity decreases continuously with age, both for girls and boys. For example, among boys aged 14 to 17, 16% still meet the WHO guidelines, while among girls, only 7.5% do [8]. An accelerometer-based study revealed that children who attend all-day schools have on average 20 min less physical activity time than half-day school children [9]. This in turn suggests that current all-day school concepts do not offer pupils as many opportunities for physical activity as other environments.

Urs Illi [10] was one of the first to recognise the important role of schools more than 30 years ago. The basic idea of his concept called Moving School was to reduce the negative consequences of sitting too long in chairs that are not adaptable to individual student needs. This concept was well received in Germany as in many other central European countries and was further extended into a holistic view taking into account all areas of the school including lesson hours, afternoon care and lesson breaks. Publications on this topic reached a peak around the turn of the millennium [11]. Among the numerous publications, however, evaluation studies that take a holistic view of Moving School, i.e., that consider the effect and interaction of several building blocks in the various areas of everyday school life, are still the exception, especially for secondary schools. Moreover, in the few that exist, only the content of the concept has been evaluated (e.g., [11,12,13,14,15]). So far, there are no current studies that show to what extent the concept of Moving School has arrived in the theoretical specifications for higher secondary schools and what of it is actually implemented in practice.

This is a particular desideratum in research because schools have not yet managed to design the framework conditions for students in such a way that at least the minimum amount of physical activity advised by the WHO is achieved, and the negative effects of physical inactivity are mitigated [8]. Due to the increased time children and adolescents spend at school, there are very good opportunities to integrate concepts such as that of Moving School into the school day. In addition, school is the place to reach all children and adolescents. Bavarian high schools alone reported almost 320,000 students in the 2021/2022 school year [16]. The aim of the present study is to provide an overview of the current German-language literature on Moving School by means of a heuristic analysis and, in particular, to bundle the core building blocks of the concept into a new scheme by means of a critical analysis of the same. This will provide the theoretical and methodological framework for the following empirical investigation.

Second, the extent to which the core building blocks of a Moving School are taken into account in the school curriculum, in published school programmes and on school homepages will be discussed. The school curriculum has always been a central management tool of school and lesson planning. Therefore, it is an indicator of the extent to which higher secondary schools in Bavaria are adapted to the requirements of the reality of pupils’ lives in the area of movement, play and sport [17,18]. The school programme, which has an internal and external effect at the same time, has a central role in school development [19]. It presents the profile of the individual school and informs about special projects in all subject areas. In the process of school development, school programmes can fulfil a dual function in equal measure, namely, pedagogical quality assurance and development [11,20]. In addition, it usually presents the goals of school development [21,22], which gives it, among other things, the role of an accountability instrument by which parents and extracurricular partners are informed about the school’s activities [23]. In recent years, a new understanding of management has developed in the field of education, in which the individual school has been understood as a pedagogical unit of action, the central importance of professionally acting teachers for the quality of a school has been recognised, and the necessary room to manoeuvre has been successively granted [19,22]. In this process, the school programme of the individual school gained and continues to gain importance. The homepages were chosen as a basis for research because they play an important role in the public relations work of higher secondary schools and are a good up-to-date supplement to the school programmes, which are designed for the longer term [24].

## 2. Method

### 2.1. Heuristic Analysis of the Publications on Moving School

To build further research on and relate it to existing knowledge we use the method of narrative review as a special form of literature review [25] recognised as appropriate when a nonsystematic review is implemented [26]. The examination meets the SANRA criteria, the Scale for the Assessment of Narrative Review Articles [27]. We aimed to identify German literature that refers and discusses the concept of Moving School in order to develop a framemodel that determines the core building blocks of the Moving School concept [28]. Therefore, we conducted a search in the Surf Database of the Federal Institute of Sport Sciences using the most common German equivalent for Moving School, “Bewegte Schule”. Here the Boolean operator “AND” was used so that only hits were displayed that contain both words. All search areas such as title, keywords or abstract were included. The search was performed without further restrictions in terms of language, format, publication type, year of publication or duration.

Studies were chosen which, on the one hand, formulate a definition or a theoretical concept of a moving school and, on the other hand, contain an overarching conceptual reflection; i.e., they address at least two different practical areas in which movement can be integrated into everyday school life, e.g., Sitting in Motion or Moving Break. Conversely, publications that did not meet these criteria, e.g., because they focused specifically on one aspect of Moving School, were not included.

The theoretical concepts of these publications were compared with regard to commonalities.

From the critical layering of theoretical concepts, a new comprehensive model of Moving School with seven core building blocks is developed, which serves as the theoretical and methodological basis for the following empirical survey including the analysis of the overall curriculum and the exploration of the school programmes and homepages.

### 2.2. Curriculum Analysis

The current curriculum LehrplanPLUS for Bavarian higher secondary schools was analysed on whether it includes the core building blocks of Moving School previously determined. In each case, the combination of terms was searched for, e.g., moving and break, moving and school, etc., and the Boolean operator “AND” was used. However, since there is no mention of this concept itself or even one of the seven core building blocks throughout the curriculum, the term had to be defined more broadly. For this reason, the subject area “movement, play and sport” was chosen. Here, “sport” stands for mentions of the subject physical education as well as for the term sport in general (e.g., in the sense of doing sport, being active in sport, etc.). Likewise, “movement” subsumes the term itself as well as the creation of movement-friendly framework conditions in the sense of Moving School in all subjects and outside the classroom. Here, the individual terms (movement or game or sport) were searched for in each case and the Boolean operator “OR” was selected. As the terms movement, play and sport inevitably appear in physical education curricula, a search is made for where these terms are specifically used to integrate movement beyond this subject into everyday school life or into pupils’ lives outside school.

### 2.3. School Programmes and School Homepages

To identify whether and to which extent elements of Moving School were considered in higher secondary schools in Bavaria, we searched the school homepages and the school programmes for content indicating links to the seven core building blocks of the concept previously identified.

For this study, we analysed the 46 Technical University of Munich (TUM) reference higher secondary schools (these are higher secondary schools that cooperate voluntarily with the TUM in the areas of research and teacher training) and 54 higher secondary schools that were selected from the pool of 326 state higher secondary schools using simple random selection. Thus, almost 1/3 of the Bavarian higher secondary schools are included in the sample. Due to this high proportion and the random selection, it can be assumed that the sample reflects the characteristics of the population and is therefore representative of it [29].

Of the 100 selected schools 28 are “urban schools” (in cities >100,000 inhabitants) and 72 are “rural schools”. Under state sponsorship are 90 schools, and 10 schools are financed by non-state sponsors.

By means of a binary code, it was entered into the category scheme whether the concept Moving School itself or one or more of the seven core building blocks identified by the heuristic analysis (see below under Section 3.1) were mentioned on a homepage or in the school programme of a higher secondary school.

## 3. Results

### 3.1. Heuristic Analysis

The search for publications containing the keyword “Bewegte Schule” resulted in 761 hits. Of these 761, 21 met the selection criteria [11,12,13,30,31,32,33,34,35,36,37,38,39,40,41,42,43,44,45,46,47]. In publications by the same authors that have the same content, only one is mentioned with regard to the criteria listed in the methods section. In these 21 publications, a total of seven areas could be identified in which Moving School can be implemented and where opportunities for physical activity can be given. For the sake of simplicity, these core building blocks will be referred to as “Elements of Moving School” below. Of these, four take place during lessons (Moving Lessons, Sitting in Motion, Moving Physical Education and Movement Break), and three concern the school in general (Physical Activity Offers in Extra-Curricular School Sport (PAOEC), Moving Learning Space and Moving Break).

In Moving Lessons, the holistic nature of cognitive processes is seen as the basis for making moving an ally of learning [38]. A distinction can be made between learning with movement and learning through movement [38,48]. In learning with movement (e.g., gallery tour), movement accompanies the learning process. In learning through movement, movement is used specifically as a means to an end: pupils acquire the learning object through movement [38], whereby movement provides additional access to information [13].

Being able to choose and change sitting positions in everyday school life on furniture suitable for this purpose is what Sitting in Motion is all about. Different types of seating such as beanballs, stools, cuboids, etc., put the pupils in different sitting positions. Active–dynamic sitting with changes in the strain on the muscles leads to the vitalisation of learning [39]. In the practical implementation of Sitting in Motion, it is recommended to adapt the seating arrangement in the classroom to the work situation (e.g., U-shape, discussion circle, frontal arrangement, etc.) [11,12]. A further aim of Sitting in Motion is to give pupils a feeling for an ergonomic sitting posture.

Moving Physical Education always has a multidimensional perspective and is characterised by ambiguity and openness. The different perspectives offer the pupils individual approaches to movement. Competition and performance is one of them, but perspectives such as health, community experience or fun also have their place in Moving Physical Education. These perspectives are linked to the pupils’ motives for wanting to move. Physical education should be planned on the basis of these motives and the teaching concept should be based primarily on problem-oriented learning [35,38].

Movement Breaks during regular lessons are used to interrupt them in order to offer pupils a short time for movement. The time given in the literature for Movement Breaks varies between approx. 5 and 15 min [11]. The Movement Break can be performed individually, in partners or as a class. The teacher leads the exercises (directive method) or makes a free movement offer (non-directive method). Possible contents are songs with movement, gymnastics/yoga elements, coordination exercises, small ball games, etc. The aim of the Movement Break is to protect the pupils’ musculature from permanent one-sided strain. In addition, the Movement Break promotes the pupils’ willingness to learn and perform as the rhythmisation of the lessons increases their cognitive resilience [43].

Physical Activity Offers in Extra-Curricular School Sport (PAOEC) include all offers of a school that give pupils the opportunity to deal with challenging movement situations. The overriding goal is always to enable the pupils to find personal and social well-being through movement and to be able to organise their leisure time in a meaningful way with the inclusion of movement [12]. Such physical activity programmes can be, for example, school trips with physical activity programmes (skiing course or summer sports week), elective subjects with a physical activity focus or sports clubs with a competitive character (e.g. “Jugend trainiert für Olympia”).

The infrastructural framework of a school includes the school building design, the classroom design, the school furniture and the schoolyard design, thus referring to the entire school building and school grounds. All these sub-areas can be designed in the sense of a Moving Learning Space and should be considered in interaction with each other [11]. The design should correspond to the pupils’ need for movement and include challenges to move as the school space is not only a learning space but also a living space for the pupils [12]. In practice, a Moving Learning Space can include, for example, the ergonomic adaptation of school furniture in the sense of Moving Sitting, the provision of alternative teaching spaces (e.g., in the school garden or atrium), and the structuring of the school building and school grounds using rest and movement spaces (sitting corners, climbing walls, etc.).

Arrangements are created in Moving Breaks that offer the pupils the framework conditions to spend the time with playful movement activities during the regular recesses, which normally interrupt the morning lessons with two times of approx. 10–20 min duration. The moving break should not be a continuation of lessons with fixed instructions but should offer freedom and opportunities for self-realisation. For this purpose, sports equipment and demarcated areas (sports halls, areas of the playground, etc.) are made available for the pupils to move around. Here, too, the aim is to give the pupils the ability to act and to spend free time on their own or together with others in movement [13,40].

### 3.2. “LehrplanPLUS” Curriculum Analysis

The curriculum of higher secondary schools “LehrplanPLUS” is divided in several chapters that were subsequently analysed.

In Chapter 1, (“Educational mission”) general statements are made, for example, on the educational mission of the respective type of school; on pupil personality; on teaching principles; on competence orientation; on the tasks of the school community; and on topics relevant to educational policy, e.g., inclusion [20]. The subject area of movement, play or sport is mentioned once here in a list under the point “5.2 Members of the school community”: “The pupils experience the participation in school life as an opportunity to enrich their own lives: through participation in religious, artistic, cultural and sporting events as well as in competitions and anniversaries” (LehrplanPLUS, Chapter 1, 5.2 Members of the school community, emphasis by the authors).

Chapter 2 (“School and cross-curricular educational goals as well as everyday life skills and life economy”) compiles 15 cross-school and cross-curricular educational objectives that extend beyond the boundaries of a subject and should be encountered by pupils again and again. These goals describe corresponding subject areas to link subject lessons with interdisciplinary projects and school life. The networking of these areas should contribute to the development of students into holistically educated and competent personalities for everyday life [20] The individual areas are, for example, health promotion, cultural education, media education/digital education and transport education. Physical activity is mentioned under the area of “health” with the following wording:

“Health promotion aims at active preventive health care, addiction prevention and the development of a healthy lifestyle based on a physical, psychological, social, ecological and spiritual balance. The pupils deal with the topics of nutrition, movement, hygiene, stress/mental health, addiction/violence prevention and learn to deal with themselves in a mindful and responsible manner. Active leisure time activities and knowledge of coping strategies in stressful situations strengthen and protect the health of the pupils” [20] (emphasis by the authors).

Chapter 3 consists of 33 profiles of the subjects taught at the higher secondary schools (including a separate subject profile for the “Modern Foreign Languages”). Here, an “overview of the basic competences in all subjects of the year level which are to be built up over time” is given [20]. Each of these is in turn divided into sub-chapters, of which the fourth of these deal with “cooperation with other subjects”. In this way, the new curriculum aims to better link teaching content across them. We searched for links between teaching content and movement, play and sport and additionally for mentions of subjects connecting teaching content to physical education. Connections to physical education are made in the subject of biology, in which sport theory is mentioned as a connecting point, and in the two other artistic subjects of art and music. In art, reference is made to scenic play, which can be linked to physical education. In music, the “transfer of auditory impressions into […] physical forms of expression” [20] is mentioned.

Chapter 4 contains the year level profiles of the individual subjects. These consist of a list of basic competences that the pupils should have at the end of the respective year level. Since the terms movement, play and sport inevitably appear in physical education curricula, a search wasmade to see where these terms are specifically used to integrate movement beyond physical education into everyday school life or into the students’ lives outside of school. For this purpose, two mentions were found in all year level profiles (see Table 1).

In Chapter 5, the subject curricula present the contents of each subject separately for all grades and are thus the most specific part of the curriculum. The subject syllabi for physical education describe in grades 5, 9, 10 and 11 competencies and contents that refer to movement, play and sport outside of physical education (see Table 1).

### 3.3. Analysis of the School Programmes and Homepages

A freely accessible school programme was found for 73 out of 100 schools. The term Moving School is explicitly mentioned in 4% of the school programmes. At least one of the seven selected elements of the concept was found in 68% (50/73) of the school programmes. However, the significance must be put into perspective when looking at it more closely. Only the element PAOEC is mentioned by 45% of the school programmes. This category includes numerous activities of schools, such as electives, ski courses, open all-day schools, etc. A school that “only” offers basketball as an elective, for example, would also be listed here. This fact makes it necessary to further differentiate the group of schools that have elements of Moving School in their school programme beyond the PAOEC: 21% (15/73) of the higher secondary schools mention one element beyond PAOEC; 3% (2/73) of the higher secondary schools mention two elements beyond PAOEC; no higher secondary school offers more than two elements beyond PAOEC. With 13 mentions, the Moving Break takes a prominent position among the elements included. All other elements except the two mentioned so far—Moving Break and PAOEC—have a maximum of two mentions in a total of 73 analysed school programmes.

A similar picture emerges for the school homepages with a total sample of 100 homepages. The concept of Moving School is mentioned on the homepages of 3 out of 100 schools. The number of higher secondary schools that have in principle included elements of Moving School on their homepage, but without explicitly referring to the concept or using the special terms, is 94 out of 100, which is significantly larger than the number of schools that mention the concept in the school programmes: a percentage of 68% in the school programmes contrasts with a percentage of 94% in the homepages. However, the figure for the homepages is also relative, as it is for the school programmes: 20% mention one element of Moving School beyond PAOEC on their homepage (mostly Moving Break as single appointment with 18 mentions); 5% mention two elements of Moving School beyond PAOEC; no homepage showed three elements at the same time—as was the case with the school programmes.

The comparison of the school programmes shows that in almost all categories (the category Moving Learning Space is the only exception) urban schools have higher percentages than rural schools. With regard to the mentions on the homepages of the higher secondary schools, there is no significant difference. However, this equality is again relativized by a more differentiated view. The data show that the rural schools more often mention PAOEC as a monotheme but have significantly fewer Elements of Moving School beyond PAOEC (approx. 20% each). There are also large differences in the category of Moving Breaks. On the homepages, for example, over 35% of the urban higher secondary schools list this element of the concept. In the rural higher secondary schools, the figure is noticeably lower at around 15%. One common feature is that the majority of urban and rural higher secondary schools focus on only one element of Moving School. The sample contains 90 schools under state sponsorship and 10 schools that have private, municipal or church sponsors. One feature that only the non-state schools exhibit (20%) in the school programmes as well as on the homepages is Moving Lessons, which is not mentioned at any state higher secondary school.

In addition to the general recording of PAOEC, all-day schools that offer physical activity opportunities were also considered. Overall, 33% of the higher secondary schools surveyed mention that they offer an all-day school programme with opportunities for physical activity at school. The number of mentions of exercise- and sport-specific offerings outside of class ranges from no mention at all to twenty different offerings presented on the homepage. The majority of PAOEC offered by higher secondary schools are electives. In addition to the electives, there are also programmes such as “Jugend trainiert für Olympia” or internal school sports tournaments. Thus, the performance-oriented offers clearly predominate. Occasionally, there are also offers where the access to exercise is not necessarily given through the idea of performance, such as an offer through Outward Bound or the elective subject of relaxation.

## 4. Discussion

The aim of the study was to find out to what extent the concept of Moving School has found its way into the curricula, published school programmes and homepages of higher secondary schools in Bavaria. In a first step, based on a literature search, the core building blocks of Moving School were determined, which served as the basis for the creation of a category scheme. In a second step, the extent to which these elements of Moving School are taken into account in the school curricula, in the school programmes and on school homepages was examined.

The heuristic analysis showed that, overall, there is a large number of publications on the topic of Moving School. Nevertheless, only 21 studies formulate a definition or a theoretical concept of Moving School and contain an overarching conceptual reflection; i.e., they address at least two different practical areas in which movement can be integrated into everyday school life. This can be seen as a desideratum in the research on Moving School. A uniform concept for Moving School has not yet been developed. Based on the systematic literature search, seven elements were identified in which Moving School can be implemented and where opportunities for physical activity can be given. Of these, four take place during lessons (Moving Lessons, Sitting in Motion, Moving Physical Education and Movement Break) and three concern the school in general (Physical Activity Offers in Extra-Curricular School Sport (PAOEC), Moving Learning Space and Moving Break). These core elements build the basis for the following evaluation of the implementation of the concept of Moving School in Bavarian higher secondary schools.

The analysis of the Bavarian curriculum for higher secondary schools LehrplanPLUS revealed that, in the overall description of the educational mission of the schools (Chapter 1), “sporting events and […] competitions” [20] is mentioned as the only reference to the entire subject area of movement, play and sport, even though sporting competitions are coming under increasing criticism in the school sector [49]. There is a need for programmes in addition to general physical education that integrate exercise into the school day, especially to encourage children and youth who are less active in sports during their free time [50]. For this reason, it is important to get away from this one-track view of movement (movement = sport = performance measurement). Furthermore, movement, play and sport are not given their own subject area regarding school and cross-curricular educational goals (Chapter 2), even though they can be integrated into every regular lesson and into school projects and are important during breaks. Adding movement, play and sport during school hours could thus make a decisive contribution to integrating movement into the everyday life of the pupils. Chapter 3, with its sub-item “Cooperation with other subjects” [20], contains an element to link subject areas and knowledge with each other. However, the new introduction of the chapter misses its mark in the area of movement, play and sport when only 3 of the 29 subjects examined make a reference to physical education, and among these three, the subject of biology only mentions the integration of sport theory. But, in fact, there would be opportunities for cooperation between physical education and other subjects in all grades. Concrete connecting contents in grade 6 curricula are, for example, made in the subject of history, “Greek/Roman antiquity”, and in the subject of physical education, “fair play idea, cultural differences as enrichment in physical education”, e.g., with a project on the “Re-enactment of the Ancient Olympic Games”. Overall, the year level profiles and the subject curriculum of the grades describe physical education as an isolated school area and show little possibility of how the subject could programmatically integrate movement, play, and sport into school life beyond physical education. Other approaches in sport pedagogical research go even further and call for a system of interdisciplinary moving learning, in which movement is not understood exclusively as a method [51].

Freely accessible school programmes were found on the internet for 73 of the 100 higher secondary schools at the time of the study. In the 73 school programmes published on the internet, the term Moving School was found three times, 50 of the 73 schools mention at least one element of Moving School in the school programme, and 23 schools did not have any physical activity or sports programmes in their school programme. Most frequently, elements of PAOEC such as “Jugend trainiert für Olympia” carried out with school teams or school skiing courses are referred to in the school programme. For pupils, it is pedagogically valuable that school teams are run and trained at schools; however, an activation of a larger number of pupils cannot be achieved in this way, and therefore, further offers are needed. The extent to which the sports offerings are carried out by qualified sports teachers or by the staff of the all-day schools remains open, as does the content of these offerings. No mention is made of cooperation with sports clubs that could, for example, provide qualified exercise instructors. Beyond PAOEC, only a very small proportion of schools have integrated movement, play and sport elements into the theoretical framework, and moreover, they usually only implement one or at most two elements. The area of “moving break” is apparently where the most is already happening in the schools. However, other elements such as “Sitting in Motion”, “Movement Breaks” or “Moving Lessons”, which could take place in the classroom and lead to positive effects on students’ health [52,53], are not mentioned more than twice overall.

The picture is similar for the school homepages. The number of mentions of Moving School is three. Although many schools mention elements of Moving School, elements other than PAOEC are relatively rare. In line with the school programmes, the most common element offered is Moving Break, and further Moving School elements (Sitting in Motion, etc.) that take place in the classroom are very rarely mentioned. Comparing state and non-state schools, the lead of the non-state schools is clearly noticeable in almost all areas. Based on the small sample size, it can only be assumed that the state schools have some catching up to do in this area compared to the non-state schools.

Overall, it can be concluded that the understanding of Moving School as a comprehensive holistic concept is not present in the examined higher secondary schools in Bavaria. By implementing only one element of Moving School, the concept is neither understood nor implemented as a holistic idea. The goal of incorporating movement into the children’s everyday life in a sustainable way does not seem to be achievable with this. Currently, competition-oriented sport or the teaching of sports still dominates at higher secondary schools. Whether this is still the right way to go in times of increasingly underachieving sport students at schools remains highly questionable. Therefore, different approaches to sport and exercise should be offered. Compared to the SPRINT study, the last major study in this area in Germany, there are clear differences: the head teacher survey at that time showed that Moving School had been introduced at every fourth school and that as many as 27% of the schools surveyed were implementing the Moving Break [54]. However, the comparison of the data also shows that the implementation of the Moving Break as a singular component was already predominant at that time. This shows that there is a need for clarification regarding the uniform implementation of the concept.

A concrete starting point for the better integration of Moving School could be the training of the staff. The FiBSS project at the University of Paderborn is currently investigating the extent to which this affects the quality and quantity of physical activity offerings in all-day classes [55]. In addition, factors influencing the students’ enjoyment of movement are identified. The study is based on the same empirical and societal developments as this one: the current alarming findings on physical activity status, the proven positive correlations between learning and development, and the increasing number of children and adolescents cared for in all-day school settings. Although the study was conducted exclusively at elementary schools, important conclusions can certainly be drawn from the results for the topic of Moving School at all types of schools.

Nevertheless, in other federal states of Germany, the concept of Moving School is seen as a whole, and its implementation is promoted in various ways. In Baden-Württemberg, Sachsen and Hessen, for example, there is the possibility for schools to be certified as “Secondary School with a Movement Education Focus/Moving School/Health-Promoting School (Weiterführende Schule mit bewegungserzieherischem Schwerpunkt/ Gesundheitsfördernde Schule/Bewegte Schule)” [56,57,58]. The certificates are linked to conditions that are based on a holistic view of the concept and that should cover all areas of school life and have an effect on the leisure time behaviour of the pupils.

The limitation of this study results on the one hand from the chosen genre of “narrative review” itself. Narrative reviews are far less systematic than other reviews (e.g., systematic or scoping review) because they do not follow a standardized methodology and are more susceptible to selection bias [28]. The second reason is that the empirical study is based not only on the curricula but also on the published school programmes and school homepages of the higher secondary schools. In Bavaria, every higher secondary school is obliged to develop a school program, but the higher secondary schools are also free and responsible for the content. This also applies to the school homepages. Thus, there is a limitation on the reliability of this approach as, in individual cases, it may be that elements of the Moving School are mentioned on the homepage or in the school program but are not implemented in practice or vice versa.

## 5. Conclusions

Overall, the study shows that, in the German literature on Moving School, descriptions of the individual elements of the concept predominate. A holistic approach with an overarching theoretical basis that includes all areas of school is the exception so far. Across the scientific publications considered, seven core elements could be identified, which in interaction cover the entire school life. So far, the concept of Moving School has had very little impact on the curriculum design for higher secondary schools in Bavaria. In the new LehrplanPLUS, the idea is not mentioned either as a whole or in its partial aspects. Even the broader area of “movement, play and sport” is hardly integrated into the curriculum beyond the subject curricula of physical education. In one-fifth of the examined school programmes and school homepages, at least one element of Moving School except PAOEC is mentioned. Again, the integration of several elements of the concept is largely lacking at the schools studied, so only partial areas of school life have been covered in each case. It is clear, however, that at the schools themselves the concept has so far been incorporated more strongly into the theoretical specifications for school development than by the designers of the curriculum, even if this is performed at a low level. In a next step, a questionnaire study to better capture the perceptions of schoolteachers regarding the current practical implementation, as well as the conditions needed for the Moving School concept to gain greater acceptance in higher secondary schools, is needed. Considering the limitations of the study, a systematic empirical survey of the research field would be desirable for the future, on the basis of which concrete guiding principles for positively influencing school practices can be developed. A concrete starting point for a better integration of Moving School into the whole school day could be the training of the entire teaching staff.

## Figures and Tables

**Table 1 children-10-01395-t001:** Content of the year level profiles of physical education that refer to movement, play and sport outside of physical education described in the curriculum LehrplanPLUS.

Chapter 4	Grade 5	“Pupils Move Safely in the Familiar School Environment”
	Grade 11	“The pupils put fairness and teamwork into practice also beyond the physical education lessons and organise small sports competitions independently”
Chapter 5	Subject syllabus of physical education Grade 5	“The students… move and orientate themselves safely in their new school environment”
	Subject syllabus of physical education Grade 9	“The pupils assess local extracurricular sports offers with regard to their own inclinations and abilities, also with regard to the lifelong practice of sport and its health-promoting function. Inform themselves purposefully about career opportunities in the field of sport and report on them. […] Content related to the competences: Reference to local sports offers, e.g., clubs, commercial offers”
	Subject syllabus of physical education Grade 10	“The pupils justify the contribution of sport to a meaningful, health-promoting and -maintaining leisure time and are able to independently implement the skills and abilities acquired in school sport in their leisure time. […] Content related to the competences: Positive aspects of recreational sports, e.g., training the cardiovascular system, increasing mobility and joy of life. Leisure-oriented sports, e.g., swimming, beach volleyball, badminton”
	Subject syllabus of physical education Grade 11	“The pupils assess body ideals, which are questionable with regard to health as well as changing social behaviours and critically engage with their relationship between physical activity time and media consumption. organise small sports competitions independently and participate in school sports events. experience and evaluate sporting leisure activities with regard to their lifelong and health-promoting significance. […] Content related to the competences: Aspects of recreational sport, e.g., psychological, physical and social well-being, reflection on one’s own sporting profile Determination and critical reflection of the individual physical activity time of a week, also in comparison to the amount of time spent using media”

## Data Availability

The data presented in this study are available on request from the corresponding author.

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
