# Peer review of "The Concept of Moving School and Its Practical Implementation in Bavarian Higher Secondary Schools"

_children, 2023, doi:10.3390/children10081395_

Round 1

Reviewer 1 Report

Thank you for submitting your manuscript. After careful evaluation, I believe that this manuscript needs some major revisions, but has the potential to be published in the future. I have found the paper well written and organized correctly. I would like to provide a couple of comments:

Abstract: Please consider revising the abstract after incorporating the suggested revisions throughout the manuscript.

The background needs a common thread and be more focused on the subject. What concepts will answer your aim? Clarify and develop the rationale (why is this study important) and clarify the aim. Please emphasise the real value of the research.

Literature review is extensive and provides solid grounds for the research study. However it would be worthwhile to supplement the existing body of literature with some recent sources. The literature review should be better linked to the empirical part.

Methods section needs to be more precisely described, especially in terms of study design. Database and article selection process (e.g. language, full text etc). The keyword search should clarify what expressions and boolean operators were used (cirruculum anyalysis).

Do schools have any obligations regarding the information they provide on their website related to the topic?

I would suggest that there should be more specific references in the discussion section. Your results should be compared with the findings of other studies in topic, or similar topic.

The article must definitely include its limitations and constraints. It is completely missing from the article.

Please comment on the representativeness of the data.

What is the generalizability of your findings?

I recommend incorporating additional suggestions based on the findings into the Conclusion section of the study.

Minor editing of English language required.

Author Response

Dear Reviewers,

Thank you very much for the positive and constructive feedback on our paper and the invitation to resubmit it. We believe that the quality of the manuscript has now increased due to your valuable input!

We have revised the paper according to your comments, requests, and suggestions. On the following pages, you can find each comment and our corresponding answer to it.

Reviewer #1:

  1. The background needs a common thread and be more focused on the subject. What concepts will answer your aim? Clarify and develop the rationale (why is this study important) and clarify the aim. Please emphasise the real value of the research.

We tried to make the goal of the study clear for the first time in one sentence on pages 2 (line 97):

„The aim of the present study is to provide an overview of the current German-language literature on Moving School by means of a heuristic analysis and, in particular, to bundle the core building blocks of the concept into a new scheme by means of a critical analysis of the same. This will provide the theoretical and methodological framework for the following empirical investigation.“

The repeated presentation of the objectives, such as on page four and page nine, attempts to provide the reader with a common thread to guide them through the essay and better connect the individual parts.

„From the critical layering of theoretical concepts, a new comprehensive model of Moving School with seven core building blocks is developed, which serves as the theo-retical and methodological basis for the following empirical survey including the anal-ysis of the overall curriculum and the exploration of the school programmes and homepages.“ (page 4, line 159)

“The aim of the study was to find out to what extent the concept of Moving School has found its way into the curriculum, published school programmes and homepages of higher secondary schools in Bavaria. In a first step, based on a literature search the core building blocks of the concept Moving School were determined, which served as the basis for the creation of a category scheme. In a second step, the extent to which these elements of Moving School are taken into account in the school curriculum, in the school programmes and on school homepages was examined.” (page 9, line 386)

  1. However it would be worthwhile to supplement the existing body of literature with some recent sources.

The basic literature in the introduction has been replaced or supplemented in some places by more recent studies. For example, on page 3 (line 108) KMK 2001 was replaced by Wacker & Hübner 2019 and Stibbe 2016. Chau et al. 2010 was replaced by Wilhite et al. 2023 (page 2, line 49). In addition, the following studies were added: Altrichter, 2019 and Steffens & Haenisch, 2019, which look at the school programm as a research base and Tubin & Klein, 2007 for school homepages (page 3, line 124 and line 128).

However, one of the purposes of the introduction is to show the historical progression of the development of the Moving School concept from the pioneering work by Urs Illi to the holistic concept. Therefore, the older sources on page two are also of importance.

In the heuristic investigation, studies were not specifically excluded, but there were no more recent studies given the search criteria mentioned. This is also an insight that can be gained from this.

  1. The literature review should be better linked to the empirical part.

In order to link the literature analysis more closely to the empirical investigation, new text passages were inserted in two places:

Page 4, line 159:

„From the critical layering of theoretical concepts, a new comprehensive model of Moving School with seven core building blocks is developed, which serves as the theo-retical and methodological basis for the following empirical survey including the anal-ysis of the overall curriculum and the exploration of the school programmes and homepages.“

Page 2, line 97:

„The aim of the present study is to provide an overview of the current German-language literature on Moving School by means of a heuristic analysis and, in particular, to bundle the core building blocks of the concept into a new scheme by means of a critical analysis of the same. This will provide the theoretical and methodological framework for the following empirical investigation.“

  1. Methods section needs to be more precisely described, especially in terms of study design. Database and article selection process (e.g. language, full text etc). The keyword search should clarify what expressions and boolean operators were used (cirruculum anyalysis).

The method and type of study were defined and described in more detail (page 3, line 132):

„To build further research on and relate it to existing knowledge we use the method of narrative review as a special form of literature review (Snyder, 2019) recognized as appropriate when a nonsystematic review is implemented (Gregory & Denniss, 2018). The examination meets the SANRA criteria, the Scale for the Assessment of Narrative Review Articles (Baethge, Goldbeck-Wood, & Mertens, 2019). We aimed to identify German literature that refers and discusses the concept of Moving School in order to develop a framemodel that determines the core building blocks of the Moving-School-concept (Rumrill, Philip, & Fitzgerald, 2001). Therefore, we conducted a search in the Surf Database of the Federal Institute of Sport Sciences using the most common german equivalent for Moving School “Bewegte Schule”. Here the boolean operator "AND" was used, so that only hits were displayed that contain both words. All search areas such as title, keywords or abstract were included. The search was performed without further restrictions in terms of language, format, publication type, year of publication or duration.“

Also in the method description of the curriculum analysis, the description of the procedure was provided with further details (see page 4, line 163). The building blocks of Moving School as fixed terms such as Moving Break were each searched for with the bolean operator "AND".

The extended search for the subject area of movement, play and sport was carried out with the individual terms, using the bolean operator "OR".

  1. Do schools have any obligations regarding the information they provide on their website related to the topic?

This issue is addressed in the Limitations (page 11, line 503)

„The second reason is that the empirical study is based not only on the curriculum but also on the published school programs and school homepages of the higher secondary schools. In Bavaria, every higher secondary school is obliged to develop a school pro-gram, but the higher secondary schools are free and responsible for the content. This also applies to the school homepages. Thus, there is a limitation on the reliability of this ap-proach as in individual cases, it may be that elements of the Moving School are men-tioned on the homepage or in the school program, but are not implemented in practice, or vice versa.“

  1. I would suggest that there should be more specific references in the discussion section. Your results should be compared with the findings of other studies in topic, or similar topic.

Three studies were included in the discussion and compared with our results:

Messmer & Brea, 2014 (page 10, line 433):

„Other approaches in sport pedagogical research go even further and call for a system of interdisciplinary moving learning, in which movement is not understood exclusively as a method.“

Brettschneider, et al., 2005 (page 10, line 472):

„Compared to the SPRINT study, the last major study in this area in Germany, there are clear differences: the head teacher survey at that time showed that the Moving-School had been introduced at every fourth school and that as many as 27% of the schools surveyed were implementing the Moving Break (Brettschneider, et al., 2005). However, the comparison of the data also shows that the implementation of the Moving Break as a singular component was already predominant at that time. This shows that then, as now, there is a need for clarification regarding the uniform implementation of the concept.“

Universität Paderborn, 2023 (pages 11, line 480):

„A concrete starting point for a better integration of the concept Moving School could be the training of the staff. The FiBSS project at the University of Paderborn is currently investigating the extent to which this affects the quality and quantity of physical activity offerings in all-day classes (Universität Paderborn, 2023). In addition, factors influencing the students' enjoyment of movement are identified. The study is based on the same empirical and societal developments as this one: the current alarming findings on physical activity status, the proven positive correlations between learning and development, and the increasing number of children and adolescents cared for in all-day school settings. Although the study was conducted exclusively at elementary schools, important conclusions can certainly be drawn from the results for the topic of Moving School at all types of schools.“

  1. The article must definitely include its limitations and constraints. It is completely missing from the article.

The limitations and restrictions have been added on page 11 (line 500):

„The limitation of the study results on the one hand from the chosen genre "narrative review" itself. Narrative reviews are far less systematic than other reviews (e.g., sys-tematic or scoping review) because they do not follow a standardized methodology, and are more susceptible to selection bias (Rumrill, Philip, & Fitzgerald, 2001). The second reason is that the empirical study is based not only on the curriculum but also on the published school programs and school homepages of the higher secondary schools. In Bavaria, every higher secondary school is obliged to develop a school program, but the higher secondary schools are free and responsible for the content. This also applies to the school homepages. Thus, there is a limitation on the reliability of this approach as in individual cases, it may be that elements of the Moving School are mentioned on the homepage or in the school program, but are not implemented in practice, or vice versa.“ 

  1. Please comment on the representativeness of the data.

For this purpose, a section and a literature source have been added on page 4 (line 186):

„Thus, almost 1/3 of the Bavarian higher secondary schools are included in the sample. Due to this high proportion and the random selection, it can be assumed that the sample reflects the characteristics of the population and is therefore representative of it (Fröhlich, Mayerl, Pieter, & Kemmler, 2020).“

  1. What is the generalizability of your findings?

On page two, it is now shown that about 320,000 students are taught at Bavarian higher secondary schools alone. Approximately 1/3 of all state higher secondary schools were included in the study. Therefore, it can be assumed that the results are also transferable. In addition, three further studies were included in the discussion, with whose results the present results can be compared (see also point 6).

  1. I recommend incorporating additional suggestions based on the findings into the Conclusion section of the study.

In the conclusion we have added the following sentence (page 11, line 529):

„Considering the limitations of the study, a systematic empirical survey of the research field would be desirable for the future, on the basis of which concrete guiding principles for positively influencing school practice can be developed. A concrete starting point for a better integration of the concept Moving School could be the training of the staff.“

Reviewer #2:

  1. The biggest question I have is whether your research is a systematic review, or a bibliographic review. Reading through the concepts used in your search strategy, I see that only one key concept was used. Therefore, I believe that it would not be a systematic review. Please review and modify where necessary.

For this purpose, the following new section has been added to the "method" chapter on page 3 (line 132):

„To build further research on and relate it to existing knowledge we use the method of narrative review as a special form of literature review (Snyder, 2019) recognized as appropriate when a nonsystematic review is implemented (Gregory & Denniss, 2018). The examination meets the SANRA criteria, the Scale for the Assessment of Narrative Review Articles (Baethge, Goldbeck-Wood, & Mertens, 2019). We aimed to identify German literature that refers and discusses the concept of Moving School in order to develop a framemodel that determines the core building blocks of the Moving-School-concept (Rumrill, Philip, & Fitzgerald, 2001). Therefore, we conducted a search in the Surf Database of the Federal Institute of Sport Sciences using the most common german equivalent for Moving School “Bewegte Schule”. Here the boolean operator "AND" was used, so that only hits were displayed that contain both words. All search areas such as title, keywords or abstract were included. The search was performed without further restrictions in terms of language, format, publication type, year of publication or duration.“

  1. In what was stated in the introduction, there is a lack of authors who support many of the assertions that are made. Despite this, the ideas are giving an overview of their research topic, what is known and what is not known about the subject is exposed, but it is necessary to highlight more importantly, the damages of not having this information.

The sources in the introduction were supplemented to better justify the statements about school homepages and school programs as the basis for the study. In addition, older sources were replaced by newer ones and the aim of the study was presented more clearly. For example, on page 3 (line 108) KMK 2001 was replaced by Wacker & Hübner 2019 and Stibbe 2016. Chau et al. 2010 was replaced by Wilhite et al. 2023 (page 2, line 49). In addition, the following studies were added: Altrichter, 2019 and Steffens & Haenisch, 2019, which look at the school programm as a research base and Tubin & Klein, 2007 for school homepages (page 3, line 124 and line 128).

In addition, studies that support the methodological approach with scientific evidence have also been added to the methods section (see page 3, line 132).

  1. Regarding the objectives, I do not clearly understand what their objectives are, which is ratified when reading their results, since they are not in the same line as their objectives.

See point 1 of reviewer #1.

Reviewer 2 Report

Dear All,

I find the work they do interesting, but from my point of view it is necessary to make great adjustments for it to be published.

The biggest question I have is whether your research is a systematic review, or a bibliographic review. Reading through the concepts used in your search strategy, I see that only one key concept was used. Therefore, I believe that it would not be a systematic review. Please review and modify where necessary.

In what was stated in the introduction, there is a lack of authors who support many of the assertions that are made. Despite this, the ideas are giving an overview of their research topic, what is known and what is not known about the subject is exposed, but it is necessary to highlight more importantly, the damages of not having this information.

Regarding the objectives, I do not clearly understand what their objectives are, which is ratified when reading their results, since they are not in the same line as their objectives.

By being clear about whether your review is systematic and bibliographical, I will be able to review the methodology chapter with greater certainty.

I believe that the results are correctly described, but as I have presented them to you before, they are not clear to me and they do not align with your objectives.

I hope my comments will help you to improve your manuscript.

Greetings.

Author Response

(The authors gave the same response as above.)

Round 2

Reviewer 2 Report

Dear All,

All the points I had doubts about in the first review were addressed and clarified.

The changes made have enhanced what they had at first.

happy